# Modern Techniques in Colorado Potato Beetle (*Leptinotarsa decemlineata* Say) Control and Resistance Management: History Review and Future Perspectives

**DOI:** 10.3390/insects11090581

**Published:** 2020-09-01

**Authors:** Martina Kadoić Balaško, Katarina M. Mikac, Renata Bažok, Darija Lemic

**Affiliations:** 1Department of Agricultural Zoology, Faculty of Agriculture, University of Zagreb, Svetošimunska 25, 10000 Zagreb, Croatia; rbazok@agr.hr (R.B.); dlemic@agr.hr (D.L.); 2Centre for Sustainable Ecosystem Solutions, School of Earth, Atmospheric and Life Sciences, Faculty of Science, Medicine and Health, University of Wollongong, Wollongong 2522, Australia; kmikac@uow.edu.au

**Keywords:** Colorado potato beetle, resistance problem, control strategies, GM potato, RNAi, SNPs

## Abstract

**Simple Summary:**

The Colorado potato beetle (CPB) is one of the most important potato pest worldwide. It is native to U.S. but during the 20th century it has dispersed through Europe, Asia and western China. It continues to expand in an east and southeast direction. Damages are caused by larvae and adults. Their feeding on potato plant leaves can cause complete defoliation and lead to a large yield loss. After the long period of using only chemical control measures, the emergence of resistance increased and some new and different methods come to the fore. The main focus of this review is on new approaches to the old CPB control problem. We describe the use of *Bacillus thuringiensis* and RNA interference (RNAi) as possible solutions for the future in CPB management. RNAi has proven successful in controlling many pests and shows great potential for CPB control. Better understanding of the mechanisms that affect efficiency will enable the development of this technology and boost potential of RNAi to become part of integrated plant protection in the future. We described also the possibility of using single nucleotide polymorphisms (SNPs) as a way to go deeper into our understanding of resistance and how it influences genotypes.

**Abstract:**

Colorado potato beetle, CPB (*Leptinotarsa decemlineata* Say), is one of the most important pests of the potato globally. Larvae and adults can cause complete defoliation of potato plant leaves and can lead to a large yield loss. The insect has been successfully suppressed by insecticides; however, over time, has developed resistance to insecticides from various chemical groups, and its once successful control has diminished. The number of available active chemical control substances is decreasing with the process of testing, and registering new products on the market are time-consuming and expensive, with the possibility of resistance ever present. All of these concerns have led to the search for new methods to control CPB and efficient tools to assist with the detection of resistant variants and monitoring of resistant populations. Current strategies that may aid in slowing resistance include gene silencing by RNA interference (RNAi). RNAi, besides providing an efficient tool for gene functional studies, represents a safe, efficient, and eco-friendly strategy for CPB control. Genetically modified (GM) crops that produce the toxins of *Bacillus thuringiensis* (*Bt*) have many advantages over agro-technical, mechanical, biological, and chemical measures. However, pest resistance that may occur and public acceptance of GM modified food crops are the main problems associated with *Bt* crops. Recent developments in the speed, cost, and accuracy of next generation sequencing are revolutionizing the discovery of single nucleotide polymorphisms (SNPs) and field of population genomics. There is a need for effective resistance monitoring programs that are capable of the early detection of resistance and successful implementation of integrated resistance management (IRM). The main focus of this review is on new technologies for CPB control (RNAi) and tools (SNPs) for detection of resistant CPB populations.

## 1. Introduction

### Colorado Potato Beetle—A Global Pest of Potato Production

Potato (*Solanum tuberosum* L.) is an especially important crop worldwide. According to Food and Agriculture Organization of the United Nations (FAO STAT) [1], it is the fourth most important food crop, following wheat, rice, and maize. More than 1 billion people consume potatoes as a staple, and the crop plays an increasingly important role in future global food security. At a global scale, approximately 20 million hectares are planted with an average yield of 17 tons/hectare resulting in 370 million tons valued annually at approximately US $50 billion [1]. Without crop protection, about 75% of attainable potato production would be lost to pests [2]. Oerke [3] estimated quantitative losses of potato due to insect pests to be around 34% annually. 

The Colorado potato beetle, CPB *(Leptinotarsa decemlineata* Say) is the main insect pest of potato plants [4]. According to Weber [5], its current distribution covers about 16 million km^2^ in North America, Europe, and Asia. It was first observed in the U.S. in 1811 by Thomas Nuttall [6]. The first serious damage to the potato in the U.S. was observed in 1874 in Colorado [7]. In the first several years after appearing, the CPB turned out to be a very devastating potato pest [8]. In Europe, the first CPB population was discovered in Germany in 1877, but it was successfully eradicated at that time. However, in 1922, CPB population was established in France [9], and by the end of 20th century, it spread across Europe (Figure 1), Asia, and western China. CPB continues to expand in an east and southeast direction [5]. Cong et al. [10] reports that CPB has been found in provinces in Northeast China; hence, we can say that China has become the frontier for the global CPB spread.

Damage to potato plant leaves caused by the CPB adults and larvae appears as holes of varying sizes, usually starting around the margins. The leaf blades are eaten, often leaving a skeleton of veins and petioles behind. This can result in defoliation. A single CPB during its larval stage can consume 40 cm^2^ of potato leaves [11]. Then, when the plant has been defoliated, adult CPB feed on stems and exposed tubers [6]. Defoliation of potato plants by the CPB can completely destroy potato crops and significantly decrease tuber production [12,13]. Control of this pest has proved very challenging because of its highly destructive feeding habits and its ability to adapt to a range of environment stresses [14] that would otherwise suppress other Chrysomelidae pests [15].

Current CPB management and control practices include biological control, cultural practices, and chemical treatments [9,14]. Overwhelmingly, historical and contemporary CPB control strategies have relied upon insecticides [16]. Gauthier et al. [17] stated that CPB has been credited with being largely responsible for creating the modern insecticide industry. Even though the use of insecticides resulted in a drastic reduction of CPB populations, resistance development against the active substances resulted. It is now well documented that CPB have developed resistance to most registered insecticides [18,19,20,21,22]. Currently, CPB has developed resistance to 56 different compounds (Figure 2) belonging to all major insecticide classes [23].

Given that CPB has developed resistance to all major classes of chemical insecticides, other control solutions are required. One such possible solution is genetically modified (GM) crops. In the worldwide cultivation of GM crops, cotton and maize varieties are most represented [2]. *Bacillus thuringiensis* (*Bt*), maize expressing crystalline (Cry) toxin (Cry3Bb1) that specifically targets the western corn rootworm (WCR), *Diabrotica virgifera virgifera* LeConte, (Coleoptera; Chrysomelidae) has increased rapidly since commercialization in 2003 [24]. Currently, a number of genetically modified *Bt* crop cultivars are widely used by farmers as alternatives to chemical insecticides for control of economically important insect pests globally (United States, Canada; India, China, Brazil, Argentina, South Africa) [2]. In 2016, the total area cultivated with GM crops globally was estimated as 185 million hectares [25]. 

There are no genetically modified potatoes in production in the European Union (EU), but through breeding programs commercial seed companies are working on mitigating the resistance of potato varieties to late blight, caused by the fungus *Phytophthora infestans* (Peronosporales; Peronosporaceae). There are five major potato-breeding companies in Europe: Kweekbedrijf Smeenge-Research, Solana, HZPC, Nijs Potatoes, and Meijer Potato [26]. Potato breeding is considerably time consuming as it takes between eight to 15 years to develop and introduce new varieties to market [26]. On the EU market, there are no commercial cultivars of potato for human consumption that show a strong level of resistance towards the CPB [27]; the cultivar Dakota Diamond has shown some level of host resistance however [28].

While genetically modified potato is not mandated in production systems globally [2], and breeding programs are yet to develop resistant cultivars it is nevertheless important to evaluate current knowledge on and modern approaches to CPB control and resistant management.

## 2. *Bacillus thuringiensis* (*Bt*) in the Fight to Control Colorado Potato Beetle

Current integrated crop management strategies for potato cultivation include combination of cultural practices, biological control, and chemical treatments [14]. As a result of CPB resistance to insecticides, and various health and environment concerns connected with pesticides, there is an increasing public demand for the reduction of pesticide use [29]. *Bacillus thuringiensis* (*Bt*) strains have been used as foliar sprays against various pests [30]. *Cry* proteins are the primarily active components of *Bt*-based microbial insecticides, which have been used as foliar sprays in agriculture for several decades [31]. *Bacillus thuringiensis* var. *tenebrionis* (B. t. t.) produces a parasporal crystal protein, Cry3A, which is displaying insecticidal properties towards CPB. This protein is characterized by its high unit activity and specificity for certain coleopteran insect pests including CPB [31]. The advantage of *Bt* insecticides is that they are generally not harmful to humans, non-target wildlife, or beneficial arthropods. The unique mode of action and selectivity make *Bt* an important alternative to conventional chemical insecticides in many integrated pest management (IPM) programs. However, the use of *Bt* sprays provides only limited plant protection as the toxins are photosensitive and degrade quickly compared to most other chemical insecticides [32]. Moreover, the use of *Bt* sprays for pest control raises concerns about the potential for accelerated resistance development to *Bt* [33,34].

*Bt*-derived Cry genes are also widely used to generate transgenic plants resistant to insects [35]. The first genetically modified potato cultivars, expressing the Cry3A toxin, were introduced in 1995 [36]. One of the first experiments occurred in which the Cry3A protein was inserted into potato plants by Perlak et al. [31]. By the insertion of a Cry3A gene, Russet Burbank potato plants were genetically improved to resist insect attack and damage (Figure 3). Results showed that the damage by all insect stages in the laboratory and also at multiple field locations was significantly reduced. Further analyses showed that GM-potatoes were the same quality in terms of agronomic characteristics including taste in comparison with the standard or non-GM Russet Burbank potatoes. The GM variety for human food was commercially available in the USA from 1996 until 2001, and during that time, ensured good control against the CPB [16]. However, because of complications connected with planting GM potatoes, new insecticide compounds, and rejection of the public, GM potato did not sustain long on the market. “Amflora”, is currently the only GM potato variety grown commercially and it is approved only for industrial use and animal feed [2].

### 2.1. Bt Potato Development

Modified Cry3Aa1 gene has been used to enhance protection of the Russet Burbank potato variety against the CPB [31,37]. Another Cry3 gene, Cry3Ca1, was found to be effective against CPB and was engineered for enhanced insecticidal activity [38] as well as Cry genes for Cry1 [39] and Cry3Bb1 [40].

Reed et al. [41] carried out a two-year field study to evaluate the efficacy of *Bt* potatoes (NewLeaf™, which expresses the insecticidal protein Cry3A) and conventional insecticide spray programs against CPB and their impact on non-target arthropods in a potato agro-ecosystem. There were six control regimes used in the experiment. Data generated showed that NewLeaf™ potato plants had greater efficiency in suppressing populations of CPB in comparison with early- and mid-season applications of systemic insecticides (phorate and disulfoton), bi-weekly applications of permethrin and weekly sprays of a microbial *Bt*-based formulation containing Cry3Aa. Importantly, the experiment showed that there was no significant difference on the abundance of beneficial predators or secondary potato pests among conventional potato plants not treated with any insecticides, the effective control of CPB by NewLeaf™ potato plants or weekly sprays of a *Bt*-based formulation. These findings are not surprising because the Cry3Aa protein is highly selective in its activity, affecting only Coleoptera (such as CPB) in the family Chrysomelidae [42]. Transgenic *Bt* potato and *Bt*-based microbial formulations are compatible with the development of integrated pest management (IPM). However, re-introduction of GM potatoes awaits changes in consumer preferences [16].

### 2.2. Why Bt Potato Did Not Sustain on Market

Resistance problems in the U.S. in the early 1990s reached critical levels [9] and growers in some potato-producing regions completely exhausted their chemical control options. In 1995, Monsanto introduced the NewLeaf™ potato variety to market, which was their first genetically modified crop. The use of NewLeaf™ potatoes led to a significant reduction in pesticide use and cost savings for growers [43]. However, there were concerns with NewLeaf™ potatoes. That is, CPB may also develop resistance to the *Bt* endotoxin because of its constant presence in the transgenic crop. Resistance to *Bt* toxins can emerge in CPB under high levels of *Bt* endotoxin stress [44].

Hoy [45] developed resistance management strategy, which include five main steps to avoid resistance development to the Cry3A protein. This strategy includes combining and switching varieties of potato during the planting operation. All potato growers needed to plant non-transformed potatoes along planting NewLeaf™ potatoes to reduce the potential for development of resistance. This was a complication that many potato growers were not used to and one of the factors against planting NewLeaf™. One more factor that worked against market adoption was the introduction of a new class of insecticides. A brief period of relief in areas where the beetles had developed resistance to other chemicals came with the use of neonicotinoid insecticides in 1995 [46]. The neonicotinoid imidacloprid was introduced at about the same time as NewLeaf™, and offered an effective conventional pesticide alternative to producers struggling to control beetles that were becoming resistant to other insecticides [47]. However, CPB gained resistance to imidacloprid very quickly and the first cases of resistance were reported from commercial potato farms in several U.S. States in 2000s [48,49,50,51].

When the NewLeaf™ potato became interesting to the media and the public debate about the risks and benefits of biotechnology started, potato growers, and retailers had to come up with an idea about how to respond to any potential controversy. This resulted in a strategy to separate potatoes in an effort to allow customers the ‘choice’ between GM and non-GM potatoes. However, problems arose in this strategy because GM testing protocols and segregation techniques were not well-developed [46]. Finally, growers realized that the NewLeaf™ potato was not adding value to their business, also the signals from market became less certain and many decided they could not afford the risk of planting NewLeaf™ potatoes. Many growers turned their attention and hope to the new active substances on the market. After the 1999 season, potato acreage planting declined rapidly and in response to market demands, Monsanto discontinued the sale of NewLeaf™ seed in 2001 [46]. CPB did not develop resistance to NewLeaf™ potatoes; however, because of the problems discussed, production and cultivation did not continue [46].

## 3. Sources of Host-Plant Resistance

There remains a market need for potato varieties resistant to the CPB due to resistance problems, restrictions on the registration and use of plant protection products in the EU, and the fact that the number of active substances in the insecticides market is declining. Spooner and Bamberg [52] suggested host-plant resistance as one of the practical and long-term solutions for controlling CPB. Two natural insect host plant resistance mechanisms in potatoes are leptine glycoalkaloids and glandular trichomes. Balbyshev and Lorenzen [53] found that one *Solanum* spp. hybrid responded to egg masses of the CPB with a hypersensitive necrotic zone that subsequently disintegrated around the border and detached from the leaf. Their results showed detachment of CPB eggs with subsequent deposition on the ground and this can be considered a new mechanism in host-plant resistance. Lorenzen et al. [54] described a new source of host-plant resistance to the CPB in a tetraploid potato. Their resistant genotypes included low levels of leptines I and II. Results after four days showed delayed development of neonate larva and inhibited larval weight gain by 75%, relative to larval development and weight gain on susceptible genotypes. According to several authors, leptines are effective natural mechanisms of potato resistance against CPB [55]. Coombs et al. [55] combined natural leptine glycoalkaloids and glandular trichomes and engineered *Bt* Cry3A host plant mechanisms as a possibility to prevent the resistance development to *Bt* endotoxin. Their study was the first report combining natural and engineered anti-resistance management options in potato and showed promising results for effective management of CPB.

For the development of CPB resistant potato varieties, natural variation of wild potato relatives can be used as source of resistance. Materials and tools to develop CPB resistant potato varieties through classical breeding programs and GM approaches are available and should be used to make potato production more sustainable [14]. The use of natural variation could avoid the problems with public relations and regulatory issues connected to GM crops, which is still present in many countries especially in the EU [16].

## 4. New Approaches to Colorado Potato Beetle Management

### 4.1. RNA Interference (RNAi)

RNAi is a gene silencing technology that uses double stranded RNA (dsRNA) to hinder the normal gene function directly against a specific gene sequence or promoter region of messenger RNA (mRNA) [56]. RNAi is a robust tool for the suppression of CPB gene expression and to study their biological function [57]. When dsRNA is ingested by insects, the transcript of target insect gene is silenced through RNAi pathway. Silencing of certain genes may cause insect growth or developmental defects, morbidity, or mortality [58]. The most important advantage of RNAi technology is that it acts on a specific insect species, because it targets a specific gene [59], and by altering the target genes, it is possible to completely avoid resistance development. RNAi in insects has three pathways: small interfering RNA (siRNA), microRNA (miRNA), and piwi-interacting RNA (piRNA) [60]. These pathways involve different proteins and play different roles in insects. This gene silencing strategy functions well in many coleopteran insects [61]. Analysis of the gut transcriptome indicates that CPB possesses all of the RNAi-related genes, providing a genetic basis for triggering RNAi in this pest [62]. The availability of the CPB transcriptome [63] will be very helpful in this respect. Duplications of some genes involved in the RNAi pathway might explain why CPB is more sensitive to dsRNA than other insects [64].

### 4.2. RNAi in Colorado Potato Beetle Control Management

Zhu et al. [65] investigated the potential of feeding dsRNA expressed in bacteria or synthesized in vitro to CPB to control their populations. Feeding RNAi successfully triggered the silencing of five target genes tested (*actin, vATPase A, B, E, Sec23, and COPβ*). These genes were related to cellular physiological processes and silencing them can impede growth and induce mortality. This study is the first example of an effective RNAi response in insects after feeding dsRNA produced in bacteria. Zhu et al.’s [65] results suggest that the efficient induction of RNAi using bacteria to deliver dsRNA is a possible method for the management of CPB. This could be also a promising bioassay approach for genome-wide screens to identify effective target genes for use as novel RNAi-based insecticides [65]. Numerous studies demonstrated successful knockdown of target genes in dsRNA fed CPB (Table 1). Zhou et al. [66] showed feeding bacterially expressed AdoHcy hydro-lase (*SAHase*) dsRNA to CPB decreased SAHase and Krüppel homolog 1 gene (*Kr-h1*) mRNA levels, reduced juvenile hormone (*JH*) titer, and that can cause the death of larvae, and pupae, and blocked adult emergence. Another very important study in CPB showed that feeding ryanodine receptor (*RyR*) dsRNA reduced *RyR* mRNA levels in the larvae and adults, and caused a decrease in chlorantraniliprole-induced mortality confirming that *RyR* is the target site for this insecticide [67]. The xenobiotic transcription factor Cap ‘n’ collar isoform C (*CncC*), regulates the expression of multiple cytochrome P450 genes, and plays crucial roles in CPB insecticide resistance. The suppression of *CncC* by RNAi reduced imidacloprid resistance of CPB [68]. Feeding dsRNA method has been used to knockdown expression of the gene coding for P450 enzyme Shade (*shd*). A reduction in the hydroxylation of ecdysone caused delay in development and death of CPB larvae and pupae [69]. Ochoa-Campuzano et al. [70] in their study identified prohibitin, an essential protein for CPB viability, as Cry3Aa binding protein. Combination of feeding prohibitin dsRNA and treatment with Cry3Aa enhanced Cry3Aa toxin induced mortality by threefold and the time to kill was reduced. Results showed 100% mortality in five days. Although the molecular mechanisms of synergism between prohibitin RNAi and Cry3Aa toxin application are not known yet, this study proposes an interesting method of combining RNAi with toxins derived from microbes and other sources to improve the efficacy of RNAi in pest control.

In Wan et al. [71] the authors investigated two dsRNAs (dsLdp5cdh1 and dsLdp5cdh2) that were bacterially expressed and fed to CPB adults. The result showed significant decrease in CPB Ldalt mRNA abundance, flight speed, flight duration, and flight distance, and also caused adult mortality. CPB adults are proficient fliers and flight, is their primary mode of dispersal. Wan et al. [72] in their study showed that if we know that proline is the main energy source for CPB flight knocking down the Pyrroline-5-carboxylate dehydrogenase (*P5CDh*) gene can weaken flight competence, and increase adult mortality. Flight in CPB is also connected with alanine aminotransferase (*alt*). Hussain et al. [73] focused on the suppressed transcripts level of highly expressive Ecdysone receptor (*EcR*) gene of CPB using plant-mediated RNAi approach. Bioassays of transgenic plants showed 20–80% mortality of CPB instars. Larvae feeding on transgenic potato plants showed halted metamorphosis, lower body weight, and larvae were not able to shift to their next instar. These results are very encouraging to control CPB, a notorious potato pest by using an alternative, effective, and reliable non-chemical method of population control and suppression. The dsRNA targeting CPB genes could be expressed in potato plants to control this pest.

Previous attempts at introducing transgenic potato plants to control CPB were not highly successful [87]. Petek at al. [86] in their study designed dsRNA to silence the CPB mesh gene (*MESH*). They did laboratory-feeding trials to assess impacts on beetle survival and development and also a field trial to compare dsRNA sprayed potato with a spinosad-based insecticide. Results showed that dsMESH ingestion consistently and significantly impaired larval growth and decreased larval survival in laboratory feeding experiments. Results of the field trial showed that dsMESH was as effective in controlling CPB larvae as a commercial spinosad insecticide, only its activity was slower. Most recently, Gui et al. [88] used the CRIPR/Cas9 system mutagenesis studies in the CPB for the first time. The CRISPR/Cas system is an efficient genome editing technology. First results from Gui et al. [88] showed low efficiency, but this methodology could possibly lead to the development of better and environmentally friendly CPB management strategy.

### 4.3. RNAi Based Products in Wide Use

There are three possible methods for mass-production of dsRNA for pest control: (1) expression of dsRNA in plants using transgenic technologies; (2) chemical synthesis of dsRNA in factories; and (3) production of dsRNA in microorganisms (Figure 4). Zhang et al. [76] used dsRNA targeted against the Actin-Like Protein (*ACT*) gene to produce CPB resistant potato plants. The ACT gene encodes the essential cytoskeletal protein *b-actin*. Using transgenic plants that produced the dsRNA in the chloroplast genome, Zhang et al. [76] were able to show that the resulting RNAi caused 100% mortality of CPB in five days. Hence, for CPB control chloroplast transformation is a reliable and efficient delivery method [76]. Although plant-incorporated protectants (transgenic plants) are the most cost-effective way of using RNAi-based pesticide technology, their public acceptance is challenging, especially in the EU. Another possibility, again using genetically modified organisms, is the usage of transformed insect symbionts [89] or viruses expressing pesticidal RNA molecules [90]. Thus, dsRNAs application by non-transformative strategies, i.e., through spray-induced gene silencing, is currently a more realistic option of controlling CPB [91]. Petek at al. [86] showed in laboratory trials as well as in the field that spraying with insecticidal dsRNA is a highly efficient strategy for managing CPB. Future research will have to focus on formulations to improve dsRNA stability and cellular uptake. Efficiency, safety, and possible undesirable effects of dsRNA on non-target organisms is an important though understudied topic [92].

Although in the beginnings of development, RNAi technology shows great potential for application in the control of various insect pests [62]. Several difficulties still have to be overcome before the full potential in insect pest control can be exploited [76,93,94]. Prior to its exploitation for insect pest control, it is important to document the potential limiting factors, like immune reaction and fitness cost, RNAi efficiency and dsRNA degradation, and virus-encoded suppressor of RNAi factors within the development of the RNAi-based pest control strategy. Additional challenges including the lack of feasible dsRNA delivery methods in practice, low efficiency in pest control capacity, and evolution of resistance to RNAi have largely constrained the appliance of RNAi in practice. Substantial research remains to be done before the application of RNAi in field conditions becomes an effective and cost-effective protection measure. The biggest challenge will be public acceptance. The genomes of many insects, including economically important pests, are sequenced and made available publicly to better understand RNAi processes and identify new target genes. One of the most important factors is the way in which RNA molecules are introduced into insect cells. In the future RNAi could become part of integrated plant protection measures. 

## 5. Genetic Tools in Colorado Potato Beetle Management

In addition to new and effective suppression measures for CPB, there is a need for effective resistance monitoring tools that are capable of the early detection of resistance and will allow implementation of insect resistance management (IRM) strategies. Clark et al. [95] were the first to combine three DNA based genotyping techniques for the detection of mutations associated with insecticide resistance in CPB populations. They compared bi-directional PCR amplification of specific allele (bi-PASA), single-stranded conformational polymorphism (SSCP), and minisequencing to detect mutations associated with azinphos-methyl and permethrin insecticides. These authors stated that the methods could enable the precise monitoring of the resistant and susceptible allele frequencies in field population of CPB. Udalov and Benkovskaya [96] in their review summarize the population studies of CPB. Moreover, their work shows that molecular genetic methods can be used to assess the nonspecific resistance of the CPB to insecticides.

Genetic studies of CPB started with the work of Grapputo et al. [97]. They investigated the population structure and genetic variability of North American and European populations of CPBs using mtDNA and Amplified Fragment Length Polymorphism (AFLP) markers. Understanding gene flow is particularly important for CPB management given that insecticide resistance is widespread in this species. Kumar et al. [63] subjected European CPB adult and larval transcriptome samples to 454-FLX massively parallel DNA sequencing to characterize a basal set of genes from this species. Their results offer new insights into insecticide-resistance-associated genes in this species and provides a foundation for comparative studies with other species of insects. Knowledge of evolutionary changes and the total genetic diversity of a pest population can provide useful information to understand the genetic patterns associated with each stage of the pest resistance development so that management, including monitoring and control, can be tailored to suit the resistance of the pest in question [98]. 

### Single Nucleotide Polymorphism (SNPs) as Prospective Tool in CPB Resistance Management

SNPs are single base substitutions found at a single genomic locus. They are very useful for population genetic studies because of their dense and uniform distribution within genomes (Figure 5). Recently, SNPs have become an affordable and readily accessible means of generating a lot of data quickly for non-model species [99]. SNP detection has facilitated association-mapping studies in many insect species including: *Drosophila melanogaster* Meigen, 1830 [100], *D. v. virgifera* [101], *Aedes aegypti* Linnaeus, 1762 [102], *Glossina fuscipes* Wiedemann, 1830 [103], *Diatraea saccharalis* Fabricius, 1794 [104], *Phaulacridium vittatum* Sjöstedt, 1920 [105]. Schoville et al. [64] identified 1.34 million biallelic single nucleotide polymorphisms (SNPs) from pooled RNAseq datasets in CPB from Long Island. Their result showed that CPB when compared with vertebrates (e.g., ~1 per kb in humans, or ~1 per 500 bp in chickens) and other beetles (1 in 168 bp for *Dendroctonus ponderosae* Hopkins, 1902 and 1 in 176 bp for *Onthophagus taurus* Schreber, 1759) has an exceptionally high rate of polymorphism (1 variable site for every 22 base pairs of coding DNA). Given the vast number of SNPs (thousands to millions) that are easily and affordably generated in a single sequencing run, they have surpassed microsatellites as the marker of choice when understanding the population genetics of a species [106]. Genotyping of SNPs has potentially far-reaching applications in insect population genomic studies and other insects in which specific nucleotides are statistically associated with complex phenotypic traits [107]. 

Diversity Array Technology (DArT) is method for DNA polymorphism analysis, which offers a low-cost high-throughput, robust system with minimal DNA sample requirement capable of providing comprehensive genome coverage [108]. DArTseq technology is a united one-step procedure of SNP discovery and genotyping; it enables a substantial discovery of SNPs in a wide variety of non-model organisms and provides a measure of genetic divergence and diversity within the major genetic groups [109]. The use of SNPs, in non-model organisms has become an affordable and readily accessible means of generating important data on species that otherwise would have been impossible due to cost and expertise availability [99,106]. Detailed genomic data could provide an answer about genetically conditioned resistance development in insects. The use of SNPs to understand the population genetics of CPB populations on a deeper level can be explored. Such data, which investigate genome changes associated with the development of resistance, is crucial for the implementation of agricultural, food biosecurity measures and integrated pest management strategies. Through genotyping of SNPs, an understanding of the genomic structure, population differentiation, gene flow, dispersal, and adaptive potential of CPB populations will be possible. The goal of effective and economically feasible resistance management remains impossible largely without efficient and cheap diagnostic procedures for separating susceptible and resistant genotypes [95]. Using SNPs, detection and monitoring of resistant and non-resistant variants of CPB can be performed in a novel application of this genetic marker.

## 6. Conclusions

CPB is the most harmful insect of potato that causes great economic damage to potato production worldwide. The suppression of CPB in the past relied on intensive insecticide applications, which ultimately led to the development of resistance. Now, when the number of available insecticides is decreasing, especially in the EU, we need to think about new possibilities and solutions to CPB control. Using SNPs, it should be possible to detect genetic differentiation correlated with resistance development in CPB. This would allow quick detection and monitoring of resistant variants as the first step towards the implementation of anti-resistant strategies and sustainable use of pesticide against CPB. RNAi has proven successful in controlling pests and based on research to date, shows great potential for CPB control. Better understanding of the mechanisms that affect efficiency will enable the development of this technology and boost potential of RNAi to become part of integrated plant protection in the future. Although there are barriers to overcome, the newly introduced technologies and approaches can be used to solve the problem of CPB control and resistance development.

## Figures and Tables

**Figure 1 insects-11-00581-f001:**
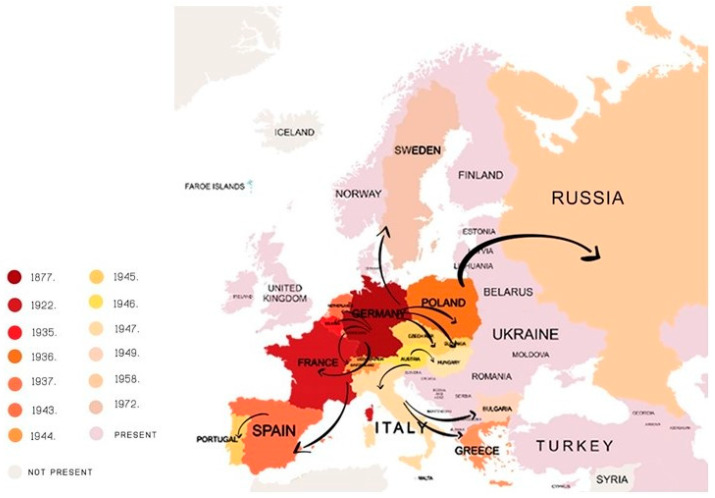
Spread of the Colorado potato beetle over Europe during the 20th century.

**Figure 2 insects-11-00581-f002:**
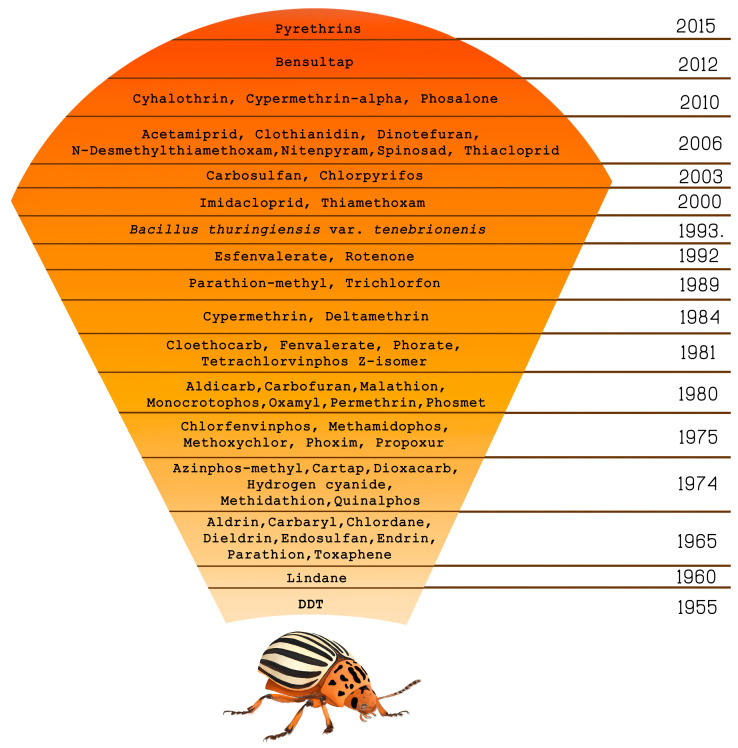
Timeline of resistance development in Colorado potato beetle.

**Figure 3 insects-11-00581-f003:**
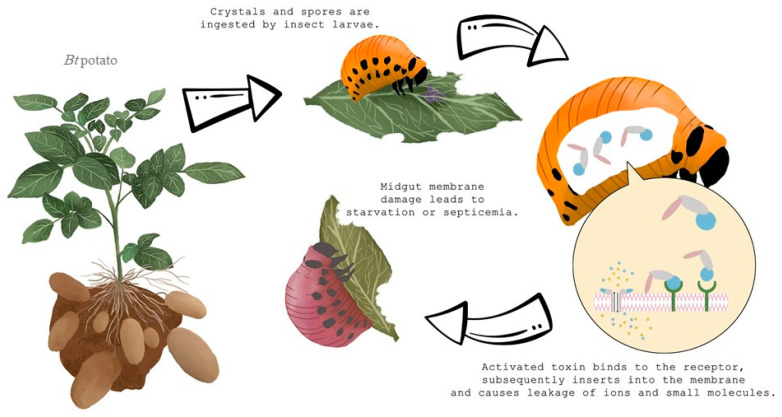
How *Bacillus thuringiensis* (*Bt)* toxin affects Colorado potato beetle larvae.

**Figure 4 insects-11-00581-f004:**
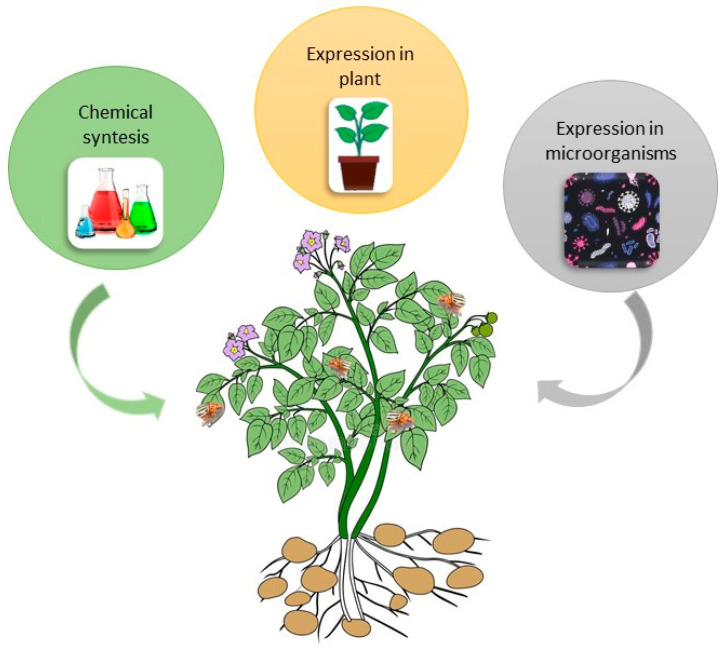
Possible methods for producing double stranded RNA (dsRNA) for pest control.

**Figure 5 insects-11-00581-f005:**
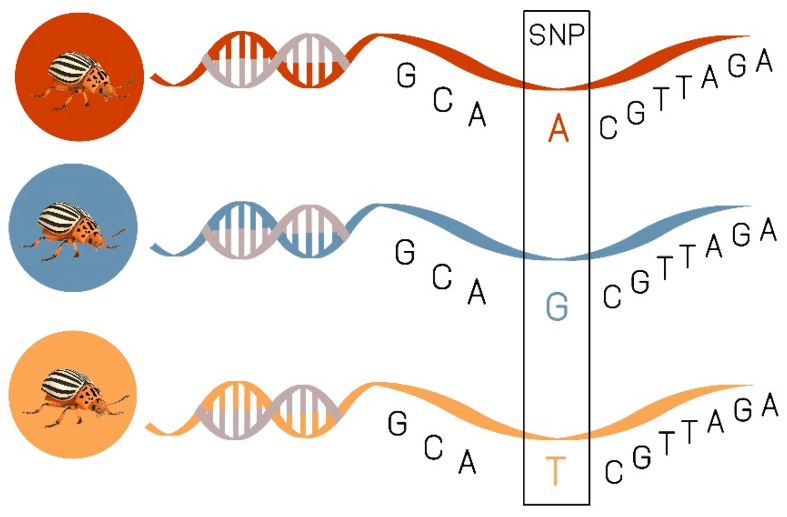
Example for single nucleotide polymorphisms (SNPs), single changes in the genetic code.

**Table 1 insects-11-00581-t001:** Review of target genes for RNA interference (RNAi)-based Colorado potato beetle control (modified from He et al. [57]).

Target Gene	Annotation	Reference
*VATPase*, *A*, *B*, *E*	Vacuolar ATP synthase subunit	[61,65]
*Sec23*	Protein transport protein *sec23*	[65]
*COPβ*	Coatomer β-subunit	[65]
*Actin*	β-Actin	[65]
*Prohibitin*	Prohibitin protein	[70]
*SAHase*	S-adenosyl-L-homocysteine hydrolase	[66]
*FTZ-F1*	Nuclear receptor type transcription factor that responses to 20-hydroxyecdysone	[74]
*shd*	Ecdysone 20-monooxygenase	[69]
*NAT1*	Nutrient amino acid transporter	[75]
*Actin*	β-Actin	[76]
*JHEH*	Juvenile hormone epoxide hydrolase	[77]
*alt*	Alanine aminotransferase	[71]
*p5cdh*	Δ1-pyrroline-5-carboxylate dehydrogenase	[72]
*HR3*	Nuclear receptor that early-late responses to 20-Hydroxyecdysone	[78]
*UAP*	Uridine diphosphate N-acetylglucosamine pyrophosphorylase	[79]
*ChS*	Chitin synthase	[80]
*TPS and TREs*	Trehalose biosynthesis and degradation	[81]
*E75*	Ecdysone-induced protein 75	[82]
*JHAMT*	Juvenile hormone acid methyltransferase	[83]
*ILP2*	Putative insulin-like peptide	[84]
*HR4*	ecdysteroidogenesis and mediates 20-hydroxyecdysone signaling during larval-pupal metamorphosis	[85]
*CncC*	Xenobiotic transcription factor	[69]
*EcR*	Ecdysone receptor	[73]
*Mesh*	gut-membrane-associated protein	[86]

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
