# Peer review of "Modern Techniques in Colorado Potato Beetle (Leptinotarsa decemlineata Say) Control and Resistance Management: History Review and Future Perspectives"

_insects, 2020, doi:10.3390/insects11090581_

Round 1

Reviewer 1 Report

#Insects-877268

Title: Modern techniques in Colorado potato beetle control and resistance management: history review and future perspectives

Brief.

The manuscript presents a review of recent techniques in CPB control and resistance management. The MS was well-written, and sections are organized connecting the subsections. However, some references were missed across the MS. Some suggestions and corrections were left in the major and minor comments in order to improve the MS.

Major comments

L46-47 - Please, a reference about the serious damage of CPB on potato crop should be provided in the sentence, such as Riley, C. V. 1875. Seventh annual report on the noxious, beneficial, and other insects of the state of Missouri. Regan & Carter, Jefferson City, Mo. and Casagrande RA [1987] The Colorado potato beetle: 125 years of mismanagement. Bulletin of the Entomological Society of America 33: 142–150.

L51 – Please, see Wang et al 2020 showed in a recent review the management methods that are used in areas at the frontier and put forward frameworks for further preventing and controlling of the spread of CPB in China (Wang et al 2020 Management of Colorado potato beetle in invasive frontier areas Journal of Integrative Agriculture Volume 19, Issue 2 Pages 360-366 https://doi.org/10.1016/S2095-3119(19)62801-7

L62-63 – Some references were missed about the control methods of CPB. Please, the references about control methods of CPB should be provided in the sentence.

L115-117 – Why were the GM-potatoes plants (Cry3A gene Russet Burbank potato plants) commercially available in the USA from 1996 until 2001? Please, a short sentence could indicate the reason of why the GM-potatoes plants remained available for only 5 years will be discussed in the subheading bellow (2.2. However, Bt potato did not gain market popularity).

L218 – As a suggestion, a short sentence regarding CRISPR/Cas9-targeted mutagenesis in the CPB in the “RNAi in Colorado potato beetle control management” section (please, see Gui, S., Nji Tizi Taning, C., Wei, D., Smagghe, G., First report on CRISPR/Cas9-targeted mutagenesis in the Colorado potato beetle, Leptinotarsa decemlineata, Journal of Insect Physiology (2020), doi: https://doi.org/10.1016/j.jinsphys.2020.104013) would be useful in addition to the RNAi study. This research provides methodology involving the use of the CRIPR/Cas9 system mutagenesis studies in the CPB. Although the efficiency was low, this methodology could be a potential method of resistance management in CPB. Thus, it would be interesting to address this issue.

L341 and L343 – The reference was indicated in the sentence, but it was inserted without the authors. Please, the authors should be inserted where was indicated in the sentence.

Minor comments

L76 – Please, the order and family name of insect species should be provided when it is cited fr the first time in the manuscript.

L83 – Please, the fungus species scientific name should be corrected in the sentence “Phytophthora”. Besides, the order and family name of fungus species should be provided in the sentence.

L271-272 – Please, just dsRNA abbreviated should be used in the sentence. This was indicated above on the line 203 and 204.

Author Response

Dear reviewer,

I would like to thank you for your time and effort in reading our article. Also, for your comments and suggestions.

REVIEWER 1

RESPONSE TO REVIEWER’S COMMENTS:

Major comments

  1. L46-47 - Please, a reference about the serious damage of CPB on potato crop should be provided in the sentence, such as Riley, C. V. 1875. Seventh annual report on the noxious, beneficial, and other insects of the state of Missouri. Regan & Carter, Jefferson City, Mo. and Casagrande RA [1987] The Colorado potato beetle: 125 years of mismanagement. Bulletin of the Entomological Society of America 33: 142–150.

Response:

References added in the text – now L48-49

  1. L51 – Please, see Wang et al 2020 showed in a recent review the management methods that are used in areas at the frontier and put forward frameworks for further preventing and controlling of the spread of CPB in China (Wang et al 2020 Management of Colorado potato beetle in invasive frontier areas Journal of Integrative Agriculture Volume 19, Issue 2 Pages 360-366 https://doi.org/10.1016/S2095-3119(19)62801-7

Response:

Reference added in the text – now L53-54

  1. L62-63 – Some references were missed about the control methods of CPB. Please, the references about control methods of CPB should be provided in the sentence.

Response:

Reference added in the text – now L65-66

  1. L115-117 – Why were the GM-potatoes plants (Cry3A gene Russet Burbank potato plants) commercially available in the USA from 1996 until 2001? Please, a short sentence could indicate the reason of why the GM-potatoes plants remained available for only 5 years will be discussed in the subheading bellow (2.2. However, Bt potato did not gain market popularity).

Response:

Short sentence about why GM potato plants remain only 5 years on market is added –now L122-123

  1. L218 – As a suggestion, a short sentence regarding CRISPR/Cas9-targeted mutagenesis in the CPB in the “RNAi in Colorado potato beetle control management” section (please, see Gui, S., Nji Tizi Taning, C., Wei, D., Smagghe, G., First report on CRISPR/Cas9-targeted mutagenesis in the Colorado potato beetle, Leptinotarsa decemlineata, Journal of Insect Physiology (2020), doi: https://doi.org/10.1016/j.jinsphys.2020.104013) would be useful in addition to the RNAi study. This research provides methodology involving the use of the CRIPR/Cas9 system mutagenesis studies in the CPB. Although the efficiency was low, this methodology could be a potential method of resistance management in CPB. Thus, it would be interesting to address this issue.

Response:

Added as suggested – now L274-277

  1. L341 and L343 – The reference was indicated in the sentence, but it was inserted without the authors. Please, the authors should be inserted where was indicated in the sentence.

Response:

Changed and added – now L350 and 352

Minor comments

  1. L76 – Please, the order and family name of insect species should be provided when it is cited fr the first time in the manuscript.

Response

The order and the family name have been added – now L79

  1. L83 – Please, the fungus species scientific name should be corrected in the sentence “Phytophthora”. Besides, the order and family name of fungus species should be provided in the sentence.

Response:

Fungus species scientific name is corrected and the order and the family name have been added – now L87

  1. L271-272 – Please, just dsRNA abbreviated should be used in the sentence. This was indicated above on the line 203 and 204.

Response:

Changed – now  L281

Reviewer 2 Report

Ln 50. Change ‘expansion’ to ‘expand.’

Ln 56. Make tuber plural (‘tubers’).

Figure 1. It would make more sense for the map just to focus on Europe. It is difficult to see which colors correspond with which dates given the current scale of the map. Africa south of Libya doesn’t need to be included, nor does the USA.

Lin. 67. Capitalize the first word in a sentence.

Ln. 76. Change ‘have’ to ‘has’ (maize has increased, not maize have increased).

Ln 97. Make spray plural (sprays).

Ln. 99. Add the reference or remove the reminder (ref).

Ln 116. Change ‘were’ to ‘was’ (the variety was available, not were available).

Ln 133. Change ‘impact’ to ‘difference.’

Ln 134. Change ‘between’ to ‘among.’

Ln 149. Remove comma (Hoy developed, not Hoy, developed).

Ln. 167. Put the word ‘adding’ in front of the word ‘value.’

Ln 178. Make solution plural (solutions).

Ln. 180. Remove the comma after ‘hybrid.’

Lns 201-203. Delete everything up to RNAi. Start the paragraph with “RNAi is a gene silencing technology…”

Ln 206. Put the word ‘and’ after the word ‘expression.’

Ln 229. Remove the word ‘that.’

Ln 237. Remove the comma after ecdysone.

Ln 249. Insert a comma after the word ‘flight.’

Lns 284-286. This sentence is hard to understand. Split into two sentences for clarity.

Ln 341. Add the references before submitting the manuscript for peer review!

Figures 3, 4 and 5 are very pretty but add very little information.

Author Response

Dear reviewer,

I would like to thank you for your time and effort in reading our article. Also, for your comments and suggestions.

REVIEWER 2

RESPONSE TO REVIEWER’S COMMENTS:

  1. Ln 50. Change ‘expansion’ to ‘expand.’

Response:

Changed – now L52

2.Ln 56. Make tuber plural (‘tubers’).

Response:

Changed – now L59

  1. Figure 1. It would make more sense for the map just to focus on Europe. It is difficult to see which colors correspond with which dates given the current scale of the map. Africa south of Libya doesn’t need to be included, nor does the USA.

Response:

We change the Figure 1 as it is suggested

  1. Lin. 67. Capitalize the first word in a sentence.

Response:

Changed – now L70

  1. Ln. 76. Change ‘have’ to ‘has’ (maize has increased, not maize have increased).

Response:

Changed – now L80

  1. Ln 97. Make spray plural (sprays).

Response:

Changed – now L102

  1. Ln. 99. Add the reference or remove the reminder (ref).

Response:

Changed – now L104

  1. Ln 116. Change ‘were’ to ‘was’ (the variety was available, not were available).

Response:

Changed – now L121

  1. Ln 133. Change ‘impact’ to ‘difference.’

Response:

Changed – now L140

  1. Ln 134. Change ‘between’ to ‘among.’

Response:

Changed – now L141

  1. Ln 149. Remove comma (Hoy developed, not Hoy, developed).

Response:

Removed – now L156

  1. Ln. 167. Put the word ‘adding’ in front of the word ‘value.’

Response:

Changed – now L174

  1. Ln 178. Make solution plural (solutions).

Response:

Changed – nowL185

  1. Ln. 180. Remove the comma after ‘hybrid.’

Response:

Removed – now L187

  1. Lns 201-203. Delete everything up to RNAi. Start the paragraph with “RNAi is a gene silencing technology…”

Response:

Deleted and changed as suggested – now L208-210

  1. Ln 206. Put the word ‘and’ after the word ‘expression.’

Response:

Changed – now L210

  1. Ln 229. Remove the word ‘that.’

Response:

Removed – nowL233

  1. Ln 237. Remove the comma after ecdysone.

Response:

Removed – nowL43

  1. Ln 249. Insert a comma after the word ‘flight.’

Response:

Inserted – nowL254

  1. Lns 284-286. This sentence is hard to understand. Split into two sentences for clarity.

Response:

The sentence is shortened and changed – now L294-295

  1. Ln 341. Add the references before submitting the manuscript for peer review!

Response:

It was a lack of concentration at the end and despite we checked the manuscript before submitting we did not see it.

22.Figures 3, 4 and 5 are very pretty but add very little information.

Response:

Considering this is a review article we thought it would be nice to complete this text with couple of figures and also to arouse the interest of the reader.

Reviewer 3 Report

Line 1 - add the latin name of cpb i the title

Line 141 - the title of the paragraph is not clear nor appropriate, please reformulate

Line 151-153 - not clear, rewrite the sentence

Line 184-187 - not clear, check the sentence

Line 202-206 - not clear check the sentence

Line 229 - Kr-h1 and JH are not previously describe, please specify these acronimous

Line 313-314 - not clear, check the sentence

Line 341 and 343 - what (references) is referred to? it is not clear if it is a definition or something missing.

Author Response

Dear reviewer,

I would like to thank you for your time and effort in reading our article. Also, for your comments and suggestions.

REVIEWER 3

RESPONSE TO REVIEWER’S COMMENTS:

  1. Line 1 - add the latin name of cpb i the title

Response:

Added as suggested

  1. Line 141 - the title of the paragraph is not clear nor appropriate, please reformulate

Response:

Modified

  1. Line 151-153 - not clear, rewrite the sentence

Response:

The sentence is rewritten – now L158-160

  1. Line 184-187 - not clear, check the sentence

Response:

Changed – now L192-194

  1. Line 202-206 - not clear check the sentence

Response:

Changed – now L208-210

  1. Line 229 - Kr-h1 and JH are not previously described, please specify these acronimous

Response:

Acronyms are specified – now L234-235

  1. Line 313-314 - not clear, check the sentence

Response:

Changed – now L321-323

  1. Line 341 and 343 - what (references) is referred to? it is not clear if it is a definition or something missing.

Response:

Corrected – now L350 and 352

Round 2

Reviewer 1 Report

#Insects-877268

Title: Modern techniques in Colorado potato beetle (Leptinotarsa decemlineata Say) control and resistance management: history review and future perspectives

Brief.

The manuscript was improved after revision. All the suggestions and corrections were incorporated into the manuscript.